# Artificial Intelligence and Its Role in Education

**Sayed Fayaz Ahmad [1,\*], Mohd. Khairil Rahmat [2], Muhammad Shujaat Mubarik [3], Muhammad Mansoor Alam [4] and Syed Irfan Hyder [5]**

[1] Department of Engineering Management, Institute of Business Management, Karachi 74900, Pakistan
[2] Centre of Research & Innovation, Universiti Kuala Lumpur, Kuala Lumpur 50250, Malaysia; mkhairil@unikl.edu.my
[3] College of Business Management, Institute of Business Management, Karachi 74900, Pakistan; shujaat.mubarik@iobm.edu.pk
[4] Faculty of Computing, Riphah International University, Islamabad 46000, Pakistan; m.mansoor@riphah.edu.pk
[5] Zia Uddin University, Karachi 75000, Pakistan; vc@zu.edu.pk
\* Correspondence: fayaz.ahmed@iobm.edu.pk

**Abstract:** The objective of this study is to explore the role of artificial intelligence applications (AIA) in education. AI applications provide the solution in many ways to the exponential rise of modern-day challenges, which create difficulties in access to education and learning. They play a significant role in forming social robots (SR), smart learning (SL), and intelligent tutoring systems (ITS) to name a few. The review indicates that the education sector should also embrace the modern methods of teaching and the necessary technology. Looking into the flow, the education sector organizations need to adopt AI technologies as a necessity of the day and education. The study needs to be tested statistically for better understanding and to make the findings more generalized in the future.

**Keywords:** artificial intelligence (AI); social robots (SR); smart learning (SL); intelligent tutoring system (ITS); education (E)

## 1. Introduction

Education is one the most essential sectors of society. It is linked with all of the other sectors and its impact on them is substantial. Due to this significance, education is indispensable for all cliques of society beyond any hurdles. For example, the challenge faced by the education sector during COVID-19 has been clear, and appeals to many researchers. But the societal challenges are not limited to such pandemics as some are always present; access to education, difficulties in reaching real classrooms, and financial issues, are some of them. There are and will be many solutions to the problems; however, this study is focused on the solution that comes from technology in the form of artificial intelligence (AI).

Artificial intelligence is changing every sector of society and the education sector is no exception. Technology has compelled many countries to implement the consumption of technology in the educational sector, such as Singapore [1], Malaysia [2], and South Korea [3]. It can be said that the future of education is coupled with technologies and their advancements. More advanced machines will open new opportunities for the education sector and will address the new challenges more effectively [4]. The AI sector has been drawing attention amongst economists [5], political analysts [6], military advisors [7], security experts [8], and education [9]. This study is focused on the AI applications: tutoring systems (TS), social robots (SR), and smart learning (SL) and their impact on education. It aims to answer the following questions:

1. What is the role of AI in education?
2. Does AI provide a solution to the difficulties associated with education?
3. Does AI benefit education?

## 2. Literature Review

### 2.1. Artificial Intelligence

The intelligence demonstrated by machines rather than humans is known as artificial intelligence (AI). Intelligence showed by humans or animals possesses consciousness and emotions while the other has no such attributes. The term AI was first used by John MsCarthy in 1955 and he defined it as "making a machine behave in ways that would be called intelligent if a human were so behaving" [10]. In 1950, Alan Turing popularized that computing machines may be thinking like humans someday [11]. He believed that in the future automated machines would make such calculations that humans would not rationally do.

Computing machines are working on binary digits and the fundamental question is how binary calculations will possess human meanings. Playing games and proving theorem are the initial endeavors for making computer machines think logically or intelligently like humans. AI often refers to machines that can perform cognitive functions in a manner associated with the human's mind, such as problem-solving and learning [12]. The device which observes its surroundings and makes decisions in a way to maximize the chance of obtaining the goals is termed an artificial agent [13]. With the passage of time and development in the areas of AI, the tasks requiring intelligence are frequently excluded from AI but have given the name artificial intelligence effect [14], because the tasks they are performing became their routine work and they became routine technology [15]. Advanced machines having the capabilities to understand human speech successfully comes in the form of AI [13]. Other machines that can also be classified as AI are those assisting in high-level strategic games and self-driving cars and the like [16]. The fundamental goal of AI research includes the representation of knowledge, reasoning, planning, learning and processing, and the capability to use objects [17]. Many types of approaches are practiced for achieving AI goals, such as statistical modeling and computational intelligence AI not only affects the field of computer science but also attracts fields of mathematics, engineering, linguistics, and many others [18].

It is clear that AI is a fast-growing field encompassing the waste boundaries of multi-discipline subjects from mathematics to engineering and from computer science to philosophy and linguistics. Due to its interdisciplinary nature, little agreement has been observed among the AI experts on its common definition and understanding [19]. As the research expands, it will have several applications in various areas. It contributes to the process of effective decision making, in games, etc., and further extends its applications into the field of education and learning. Keeping in mind the tools and services associated with AI, its presence in higher education has been noted. Interestingly many educationists are still unaware of its importance, scope, and what it consists of [20]. Considering the above issue arising from the unawareness of teachers in the use of AI in education, this research aims to further explore the AI applications in education, their scope in education, and learning.

### 2.2. Tutoring System

An intelligent system that interacts presents information and provides a test of a student's knowledge is known as an intelligent tutoring system (ITS) [21]. It is one of the sophisticated ways of information presented to the students. Like a teacher, it teaches each student according to his or her knowledge level and priorities [22]. ITS is capable of tutoring students in the following manner. Initially, it teaches and presents theory, etc. with examples. IT then asks questions from the students. It has the ability to understand the answers provided by the students and to determine their knowledge, which affects what should be presented and asked from the student. The student can also ask questions and the system has the ability to answer or solve the problems in the specific knowledge domain [23]. AI contributes significantly to the education field through ITS, which can directly impact student learning in general and in particular, the digital environment. Although it is not common yet, other digital applications have provided students such experience through the use of deep learning algorithms. ITS uses AI technology to provide

personalized teaching and feedback to students without a human teacher. Because of its ability for one-on-one curriculum, ITS has been attracting attention. Researchers are taking interest in the development of effective ITSs, which can teach different types of subjects, including equation solving, physics, mathematics, and grammar.

An intelligent tutoring system needs the following inputs:

1.  The knowledge and understanding about the course being taught, strategies for teaching, misconceptions, and possible errors.
2.  The experience acquired by the system through interaction with students. It includes the know-how of student error, student learning efforts, and their general information.
3.  The preferences or the priorities of each topic desired for student achievement level, and usability cost.
4.  The observation of the student's interaction and test results.

The outputs of ITSs are information/material presented, answers to the asked questions, the required test, and reports for parents and teachers. ITS design consists of four stages, which are assessment of needs, analysis of the cognitive task, tutor implementation, and evaluation [24]. The first stage involves learner analysis, expert or instructor consultation, and the development of student, expert, and knowledge domain. Learning goals, outcomes, curriculum structure, and task definition are the matters to be addressed here. Keeping in mind the possible behavior of students while performing a task or interacting with the system, everything needs to be planned properly. The main dimensions to be dealt with in this stage are student's problem-solving probability, time to reach a performance level, and future implementation of the knowledge [24]. The cost of the system also needs to be analyzed. The knowledge or awareness of the teachers/instructors and students also needs to be assessed as both will be the users. In the second stage, the cognitive task analysis involves the system programming with the objective of valid computational model development as per the requirement. Interviewing experts, think-aloud studies, and observation of learning behavior and teaching are the main methods for domain model development [24,25]. The primary implementation of the tutor comes in the third stage of ITS design and it involves the development of such environment that can support the learning process in an authentic manner. In the last stage, comes the evaluation processes, including pilot study testing, formative evaluation, parametric studies, and summative evaluation [24]. Some of the most common tools often used for creating ITS are ASPIRE [26], GIFT [27], the cognitive tutor authoring tools [28], AutoTutor [29], ASSISTents Builder [30] etc.

### 2.3. Social Robots

About 40 years ago, robots were introduced in classrooms and their use is increasing daily [31]. Like other intelligent systems, social robots are also intelligent machines following social behavior and interacting with humans one way or another. They have considerable use in education these days [32]. They are capable of playing multiple roles in education, such as teaching and tutoring [33]. Although modern technology in the form of AI has a significant role in education, the researchers were attracted, with divergent opinions from different areas, to debate the idea of using robots in educational settings and its values [34]. Social robots perform many tasks in the context of social interaction by evoking social responses from humans [35]. They work in collaboration with humans in many dimensions of life, and education is one of them [36].

Some argue that a human teacher is more desirable than a robot. Similarly, students also prefer a human instead, even if robots are more knowledgeable and deliver education better than a human [37]. Issues concerning the cost of technology, training, and application provide additional weight to their arguments [31]. This shows that although AI has a significant role in modern-day education, some still believe that humans are ideal. Others argue oppositely and believe that social robots have considerable and positive effects in education, and have found them to be credible sources in the context of information and knowledge. Students also see social robots capable of transmitting knowledge and

information effectively [38]. Students who have no access to educational institutes and real classrooms or cannot engage in physical classrooms have been using social robots, which allow them to interact with other students and teachers synchronously [39]. Moreover, researchers also show there to be positive effects of social robots on learning. They do not only help to provide real classrooms, such as environment, but also facilitate them to engage and communicate with other colleagues and teachers [40].

There is no doubt that social robots are helpful and productive in facilitating the students, yet making them more efficient still requires attention from the researchers. Some researchers argue that empathy and engagement are the two most important factors to be present in the human–robot interaction. It will increase bonding between the two and make the impact of a machine more effective and productive. They focus on the development of social bonding and suggest that the machines should have the capabilities to create empathic messages and social cues [41]. Robots that are physically present to deliver lectures are more effective than virtual agents. Students pay more attention to them as compared to the one on a screen, which leads to the efficiency of the learning process in the educational setting. The effects of a recorded human instructor and an animated video of a robot are analogous as the students' learning is concerned [37].

In summary, it is evident that the need for AI application in every walk of life, especially in education is irrevocable. The recent COVID pandemic is one example to clarify the needs and importance. Social robots can not only assist students by facilitating them like teachers but are also a solution to the recent challenges. In addition, it provides a learning facility for those who are not able to avail real teachers' classrooms, for example, and can provide education in remote areas. Yet there is a need to make them more social and artificially intelligent to communicate and respond in a more humanistic manner, which is perhaps one of the biggest challenges for the machines, as is a concern to researchers [42].

*2.4. Smart Learning*

Smart learning (SL) is linked with the development of smart devices based on intelligent technologies [43]. Technology is not only associated with other walks of life but also with education and can be utilized in an educational setting to help the student learning process. It has been given the name technology-enhanced learning. Technology enhances the mode of education through various means and tools for retrieving learning content [44], communication [45], evaluation [46], and expression [47] in the process of technology-enhanced learning. For example, mobile and other personal technologies have become a major source of learning by addressing the issues related to the schedules, environments, and locations, which were acting like hurdles in getting an education, and changed the traditional educational process to an advanced level free from the limitation of such challenges. Students can access the learning materials anytime, from anywhere in any environment. This was all made possible with technology development and its utilization in the education sector. This area of addressing students' education-related problems has attracted many researchers [48].

Researchers from different professions define SL differently. Some consider it to be context-aware universal orientated [49,50]. Another study considers it more focused on content and learners than its technologies, although the technological infrastructure is advanced and intelligent. It believes that the technology role is irrevocable in supporting the learning process, but the concentration should be on content and learners instead of smart devices [51]. SL combines the gains of ubiquitous learning and social learning. It is a service-oriented and learner-centric educational paradigm and is focused on both technology and learners [52,53].

It is defined as "to create intelligent environments by using smart technologies, so that smart pedagogies can be facilitated to provide personalized learning services and empower learners" [54]. It provides access to education learning sources and increases interactions between learners and instructors in an easy manner [55]. Its objective is to "improve the learning quality and student outcomes throughout the student's educational process; it

focuses on contextual, personalized, and transparent learning capable of encouraging the emergence of students' intelligence and facilitating their ability to solve problems in real environments; students are provided with personalized education where they can learn flexibly, in any place and at any time, and work collaboratively" [56,57]. It is believed that the methods of teaching and learning will be changed with the use of smart technologies. It will also impact the strategies associated with education and teaching [58]. As SL integrates the smart technology with education and learning, new pedagogies will be needed so that the learners and teachers learn how to integrate technology with their purpose [59]. Many researchers, as discussed, have studied SL with different angles, yet it should not be defined with a clear statement. It is still emerging, as is its boundaries, both technologically and content-wise [56]. Features of SL are defined as motivated, self-directed, adaptive, technology-embedded, and resource-oriented [3]. In addition, another study considers its features as formal and informal, personalized and situated, social and collaborative learning having a focus on content and application [60].

It can be summarized that SL is possible due to the use of smart technology and the artificial intelligence embodied in it. More intelligent technologies will provide more functions and utilization. It will improve the student learning outcomes. Having a focus on contextual and personalized learning, SL will enhance student's intelligence and problem-solving skills in the real environment. Students will learn without the restrictions of time and locations, in a more collaborative manner.

### 2.5. Artificial Intelligence and Education

As AI application has been playing a significant role in many sectors, the education sector has also received its attention in recent years. IT technologies and their applications are featured as one of the important development in education [61]. Education welcomes AI technologies; its application associated with learning and teaching is increasing daily. According to the Horizon report, published in 2018, AI applications will witness a 43% increase from 2018 to 2022 [61]. The report published by the same organization predicted that the increase in AI technology adoption will be even more than noted before [62]. The importance of AI in education cannot be denied and its role in the field is linked with its future [63]. Education has welcomed the entrance of AI in one way or another, yet many educationists are unaware of what it is [20]. AI is unavoidable in education and its application exists to assist educators in meeting objectives.

The question as to how AI impacts education remains. AI is a field in machine learning that consists of software capable of pattern recognition, prediction making, and learning to make a new pattern or making a decision on its own [4]. In other words, it has the know-how to respond according to the situation, which was not a program with their initial design. AI makes it happen through its rational agents, which are responsible for making a goal-oriented behavior [64]. The term, rational agent, has been previously used in game theory, economics, and decision theory, etc. has clear preferences and chooses to act in a way to obtain the highest outcome among many alternatives.

Researchers have shown that learning consists of a social exercise comprising of collaboration and interaction [65]. AI applications have three main categories in education: personal tutors, collaborative learning, and virtual reality [66]. Online collaboration needs to be moderated. Through intelligent virtual reality, students could be engaged and guided in a game-based environment of learning and reliable virtual reality, where the work of teachers, facilitators, etc. could be performed by virtual agents in remote virtual labs [67]. AI not only facilitates the process of education and learning through virtual rooms but it can also be used in assessment, especially where there are large amounts of student data [66]. It can generate a just-in-time assessment and feedback, unlike the traditional way of a stop-and-test. Through AI applications, students' learning accomplishments can be recorded and analyzed from time to time. It has algorithms for the prediction of students' progress, chances of grades to be obtained and assignment concerns with a high probability [68].

It can be summarized that AI can play a remarkable role in education. And to run with the modern advanced world, organizations in the education sector need to adopt AI technologies for education and learning. Educational organizations can implement AI according to their requirements and objectives.

## 3. Discussion

AI's role in education is through its intelligent methods for tutoring, communication, analysis, assessment and evaluation of the student or learner in addition to supervision, process control and optimization. AI technology performs all the tasks needed for a teacher and a student. The technology must be capable of communication such that the language etc. from the user side is understandable and makes sense.

AI's use in education and learning is significant. It equips the method of teaching and education with new technology and procedures. It captures researchers from diverse disciplines to study a wider range of issues related to education. It is clear AI and its role in education is cross-disciplinary and many issues beyond the scope of traditional education can be easily addressed through AI. For example, students can access learning from a location where they have no access to physical classrooms in a real environment, they merely need the required AI technology to teach them or communicate according to the teaching or learning objectives. Distance and geography are no more hurdles in education if AI is being used. Similarly, for schools and colleges, AI can make it easy to register as many students as they need regardless of their location, etc.

Unlike real teachers, AI systems communicate individually with each student and deal with them according to their need and level of understanding. Students or learners can access learning according to their understanding or knowledge level, which will be recorded in the system and will be used by the AI system to communicate next time according to that level. Unlike real classrooms, students feel at ease and can communicate with machines without any pressure and stress, which is again necessary for education and learning. Learners can register and participate without any restrictions on registration numbers. The machine will treat each student according to his/her own level of knowledge and interests. Tasks such as marking, attendance, assignment checking, etc. will be performed by the intelligent system in the absence of any influence.

In short, like every other sector, education is also influenced by modern technology and AI is one of the types. Many issues which cannot be addressed by any other mean can be addressed by the use of AI in education. Access to classrooms, content, absence of an expert teacher, and alike, are some of them. The closure of educational institutes during the COVID-19 pandemic is one of the biggest examples. AI technology and its application has assisted the sector in many ways. The need for AI is widely appreciated, and the technology has been found to be beneficial in response to the challenge.

We conclude answers to the questions designed as the aims of this paper as follows:

1. What is the role of AI in education?
2. Does AI provide a solution to the difficulties associated with education?
3. Does AI benefit education?

### 3.1. What Is the Role of AI in Education?

AI has had an enormous role in education, which has been further increased by the COVID pandemic. Perhaps the adoption and acceptance intensity of AI has been potentially amplified in the education sector. AI plays many roles in education, such as providing access, and improves communication between teachers and students. It has changed the trends of education and schooling by introducing personalization, where educating someone depends upon the knowledge level of the student, his/her speed of learning, and goals to be obtained from education or the course. Unlike the traditional manner, the learning histories of each student are analyzed continuously to assess the weaknesses and offer courses of interest and improvement.

Another significant role AI has in education is tutoring. Through its many applications and tools, AI provides tutoring, such as a chatbot or SR. It provides extra help to the students outside the classroom. AI has decreased the burden of many teachers who don't have much free time. In other words, it teaches inside and outside to cover the weaknesses of students if any, anytime from anywhere. In addition, AI has also solved the issue of timely response. It can answer repetitive and commonly asked questions in seconds and overcome the frustration of long delays. Issues of common interests and frequently asked questions are now answered by AI tools and minimize the waiting time for students and information seekers as well as the bombardment of such questions from teachers and departments. Perhaps its most amazing role is its universality. If someone has access to the internet and relevant technology, he/she can avail educational services of interest anytime from anywhere. AIAs have resolved the issues related to accessibility, health, and environment, etc. previously hindering acquiring an education. To summarize the main roles of AI in education are the automation of both administrative and academic tasks, personalization in learning, smart content, and day-night accessibility and content accessibility.

AI has also a negative role in education. Acquiring education through AIA decreases the interaction between students and teachers and does not provide the experience of the physical environment to students. This is the biggest weakness and needs further exploration.

### 3.2. Does AI Provide a Solution to the Difficulties Associated with Education?

In education, there are two types of difficulties in education, and they are academic and administrative difficulties. AIA not only assists the respective personnel and department in those tasks but also provides automatic solutions to most of them. Record keeping and admission departments are now using AI tools to minimize the burden of the tasks. Similarly, AI is also helping academic staff by carrying out their tasks effectively, such as assignment checking, exam assessment, attendance, and records. Difficulties of both types have been addressed through task automation, and intelligent tutoring.

On the student's side, the main difficulties are access to education and teaching according to the student's intelligence level. Both of them were again addressed by AIA by providing 27/7 accessibility and availability of learning material from anywhere. As AI systems assess and analyze the history, interests, and level of students, it teaches them according to their level of intelligence and interests.

### 3.3. Does AI Benefit Education?

The answer is "Yes". In addition to the role discussed in questions 1 and 2, AI benefits education in many other ways. For example, learning how to fly an airplane cannot only be obtained from books and teachers. One needs to acquire practical experience of how it feels and works. AIA, through its virtual environment, provides the required situations and the experience of how it works, etc. Inexperienced persons are at risk in coming into the actual physical environment, and many issues, such as health and safety, and phonological, can be addressed by acquiring learning through a virtual environment.

Conducting experiments in various labs are also risky and doing it personally is very difficult. AI systems in various forms can do such experiments without the risk of human loss. Such experiments are common in chemistry and physics. In the medical field, various animations and virtual images are created through AI systems, which help students learn about the function and anatomy of the human body and organs more easily than they can learn from books. Medical students now can also learn how to operate an organ of interest through the use of intelligent technology.

It is evident that not only AI and its applications help education and learning by providing access to education, introduced in many ways for delivering education, such as social robots, ITS, and SL. It also assists in carrying out academic and administrative functions and tasks; and carries the learner to a virtual reality classroom, where the skills

can be learned in the absence of risk effectively. The benefits AI has shown in the recent COVID-19 pandemic are clear and perhaps education and learning have remained intact due to AI.

### 4. Conclusions

In conclusion, AI has influenced many sectors and education is one of them. It is a contemporary method of tutoring or teaching and learning, which can address and resolve many issues related to learning. It can resolve issues, such as content accessibility, teacher deficiency where a student can learn without stress or impacting others. AI implementation and adoption is unavoidable in the education sector. AI technologies are not limited to smart learning, tutoring systems, and social robots; there are many other intelligent technologies, such as virtual facilitator, online learning environments, learning management systems, and learning analytics, which also contribute significantly to the sector.

#### 4.1. Practical Implications

1. The study provides a strong argument for the adoption and use of AIA in an educational setting.
2. It also offers education policymakers guidance about the importance and role of AIA in education and how many issues can be addressed through it.
3. It also provides educational institutions, teachers, and students with knowledge about how to use AIA, where to use it, and when to use it. Each of the parties can use the study differently according to their needs and requirements.
4. It also enlightens the educators as to how AI is changing the education world and how it can assist risky tasks.

#### 4.2. Limitations

1. The study is based on a theoretical review of the current literature to conclude answers for the questions, which were the aims of the study. There are many other AI systems playing significant roles in education, such as grading, assessment, trial and error, and virtual reality, which were not covered in this work.
2. A lack of testing the roles quantitatively to make them more generalized is another limitation.
3. The study discusses several AI applications but not extensively due to its scope in remaining limited to the management of AI.

#### 4.3. Future Work

1. As stated above, the study is based on a theoretical review of the current literature to conclude answers for the questions which were the aims of the study. There are many other AI systems playing significant roles in education, such as grading, assessment, trial and error, and virtual reality, etc. which were not covered in this work. Future work could be carried out to cover the other aspects.
2. Future work could test the roles quantitatively in order to make the research more generalized.
3. Future studies can be conducted on each AI applications in education and learning to further explore the area.

**Author Contributions:** The original draft of this research is prepared by S.F.A. and administered by M.K.R. The project is conceptualized and supervised by S.F.A. under the supervision of M.M.A., M.S.M. and S.I.H. All authors have read and agreed to the published version of the manuscript.

**Funding:** This research was funded by Univeriti Kula Lampur as a Post Doc Research Project.

**Institutional Review Board Statement:** Not applicable.

**Informed Consent Statement:** Not applicable.

**Data Availability Statement:** Not applicable.

**Acknowledgments:** We acknowledge Univeriti Kula Lampur for giving us the opportunity of Post Doctorate research and providing funding for this research. We also extend the same to Institute of Business Management for its moral support and for providing a research environment.

**Conflicts of Interest:** There is no conflict of interests.

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
