# Peer review of "Artificial Intelligence and Its Role in Education"

_sustainability, doi:10.3390/su132212902_

Round 1

Reviewer 1 Report

The authors supplemented the content of the article. I accept the response to the review. I recommend the article for printing.

Author Response

Point

The authors supplemented the content of the article. I accept the response to the review. I recommend the article for printing.

Response: Thank you very much. Your remarks mean a lot to us.

Reviewer 2 Report

Article is of interest, but seems to be a draft

Author Response

Article is of interest, but seems to be a draft

Response: Thank you very much. Your remarks means a lot for us.

The paper is written in Narrative Form. Literature Review itself is being used as Research Methodology in this paper as suggested by Hannah Snyder, Literature review as a research methodology: An overview and guidelines, Journal of Business Research, Volume 104, 2019, Pages 333-339, ISSN 0148-2963, https://doi.org/10.1016/j.jbusres.2019.07.039. 

Such types of papers are common where there is less knowledge about the problem at hand and to explore the area further. Examples of such studies are:

  1. Konrad Szocik, Mars, Human Nature And The Evolution Of The Psyche, JBIS Vol-68, 2015
  2. Konrad Szocik, Multi-Level Challenges in a Long-Term Human Space Program. The Case of Manned Mission to Mars Studia Humana. Volume 7:2 (2018), pp. 24—30, DOI: 10.2478/sh-2018-0008

Empirical confirmation of research is proposed in the future research work.

Reviewer 3 Report

The current content and structure of the study is more representative. However, doubts remain about the empirical confirmation of the relevance of research in the field of education.

Author Response

Response to Reviewer 3 Comments

The current content and structure of the study is more representative. However, doubts remain about the empirical confirmation of the relevance of research in the field of education.

Response:

 Thank you very much for praising our work as more representative. It further increased our courage and enthusiasm in researches.

The paper is written in Narrative Form. Literature Review itself is being used as Research Methodology in this paper as suggested by Hannah Snyder, Literature review as a research methodology: An overview and guidelines, Journal of Business Research, Volume 104, 2019, Pages 333-339, ISSN 0148-2963, https://doi.org/10.1016/j.jbusres.2019.07.039. 

Such types of papers are common where there is less knowledge about the problem at hand and to explore the area further. Examples of such studies are:

  1. Konrad Szocik, Mars, Human Nature And The Evolution Of The Psyche, JBIS Vol-68, 2015
  2. Konrad Szocik, Multi-Level Challenges in a Long-Term Human Space Program. The Case of Manned Mission to Mars Studia Humana. Volume 7:2 (2018), pp. 24—30, DOI: 10.2478/sh-2018-0008

We agree with the reviewer that empirical confirmation of research, makes the study more acceptable and generalized; and from this comment, future research work is proposed.

   Konrad Szocik, Mars, Human Nature And The Evolution Of The Psyche, JBIS Vol-68, 2015 and Konrad Szocik Multi-Level Challenges in a Long-Term Human Space Program. The Case of Manned Mission to Mars Studia Humana. Volume 7:2 (2018), pp. 24—30, DOI: 10.2478/sh-2018-0008

Round 2

Reviewer 2 Report

The article is not much improved,

Just the conclusions are noe in a more appropriate format

This manuscript is a resubmission of an earlier submission. The following is a list of the peer review reports and author responses from that submission.

Round 1

Reviewer 1 Report

The study offers a review of the literature on long-known facts about the use of digital technologies in education.
The research does not present the author's position, development or own methodology / technology. The article is a purely theoretical review of the literature without an emphasis on practical application.
Authors should present their work, rather than list the work of other researchers.

Reviewer 2 Report

Ian article of interest

I will suggest another round of proofreading and also to check references 

Conclusions nad further recommendations can be imoroved

Reviewer 3 Report

The subject of the article is interesting, but article needs to be supplemented with research results on the effectiveness of SL, SR, ITS in education, and if they are lacking - the research directions should be indicated.